# Polarization-driven band topology evolution in twisted MoTe₂ and WSe₂

Xiao-Wei Zhang [1], Chong Wang [1], Xiaoyu Liu[1], Yueyao Fan [1], Ting Cao [1] ✉ & Di Xiao [1,2] ✉

Motivated by recent experimental observations of opposite Chern numbers in R-type twisted MoTe₂ and WSe₂ homobilayers, we perform large-scale density-functional-theory calculations with machine learning force fields to investigate moiré band topology across a range of twist angles in both materials. We find that the Chern numbers of the moiré frontier bands change sign as a function of twist angle, and this change is driven by the competition between moiré ferroelectricity and piezoelectricity. Our large-scale calculations, enabled by machine learning methods, reveal crucial insights into interactions across different scales in twisted bilayer systems. The interplay between atomic-level relaxation effects and moiré-scale electrostatic potential variation opens new avenues for the design of intertwined topological and correlated states, including the possibility of mimicking higher Landau level physics in the absence of magnetic field.

The low-energy electronic structure of moiré superlattices can be described by Bloch electrons moving in a periodic potential that varies on the scale of the moiré period. The understanding of this moiré potential is pivotal to the realization of various topological states[1–6], including the much coveted zero-field fractional Chern insulators[7–12], recently discovered in twisted transition metal dichalcogenide (TMD) homobilayers[13–16]. Given the structural and chemical similarities among different TMDs, it is intuitive to expect that the moiré potentials of twisted TMD homobilayers, and thus the moiré band topology, would also be similar. However, recent experiments seem to suggest the contrary: at the integer hole filling of $\nu = -1$, optical and transport measurements have found opposite Chern numbers in 3.7° twisted bilayer MoTe₂ (tMoTe₂)[13–16] and 1.23° twisted bilayer WSe₂ (tWSe₂)[6].

On the theory side, discrepancies in the Chern numbers were also found by two distinct approaches used to study the moiré electronic structures. The first approach involves deriving electronic structures from small unit cells containing local stacking arrangements[17–22]. The second approach relies on density-functional theory (DFT) calculations performed on reasonably sized moiré superlattices[23–26]. Curiously, for tMoTe₂, the Chern number of the topmost spin-up (spin-down) moiré valence band is found to be −1 (+1) within the local stacking approximation[20], whereas the DFT calculation conducted on a

fully relaxed structure with a 3.89° twist yield opposite Chern numbers[25]. The latter is consistent with experimental observations. However, at smaller twist angles, the system size poses a substantial challenge to DFT calculations, and a direct comparison with experiments is currently unavailable.

In this letter, we perform large-scale DFT calculations for tMoTe₂ and tWSe₂ down to 1.25° twist angle. This is made possible by using a machine learning force field to obtain the relaxed structures, which enables a comprehensive exploration of the twist-angle dependence of the moiré lattice reconstruction. We show that the observed difference in Chern numbers is due to the twist-angle dependence of the moiré potential. Specifically, we find that as the twist angle varies, the location of the moiré potential maximum shifts from the MX stacking region to the XM stacking region (see Fig. 1 for the definition of MX and XM), causing a sign change of the Chern number. The shift of the moiré potential maximum is attributed to the competition between the in-plane piezoelectricity and the out-of-plane ferroelectricity, a mechanism associated with the broken inversion symmetry in TMDs and absent in the local stacking approximation. The large-scale calculations, enabled by machine learning methods, also reveal multiple flat bands with Chern numbers all equal to 1 in tMoTe₂ at around 2° twist, indicating the possibility of mimicking higher Landau-level

[1]Department of Materials Science and Engineering, University of Washington, Seattle, WA 98195, USA. [2]Department of Physics, University of Washington, Seattle, WA 98195, USA. ✉e-mail: tingcao@uw.edu; dixiao@uw.edu

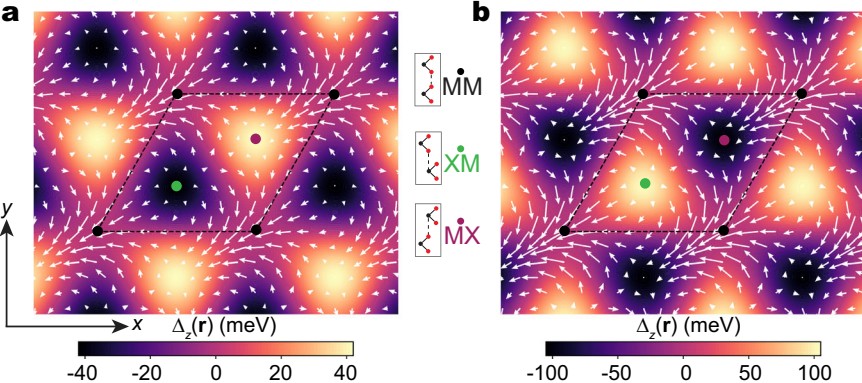

**Fig. 1 | The layer pseudospin skyrmion lattice. a** The $K$-valley layer pseudospin $\boldsymbol{\Delta}(\mathbf{r})$ skyrmion lattice, using parameters from the local stacking approximation[20]. The color denotes $\Delta_z(\mathbf{r})$ and the arrow denotes $\Delta_{x,y}(\mathbf{r})$. The black, green, and purple dots denote the three high-symmetry local stacking sites, MM, XM, and MX, respectively. **b** Similar to (**a**) but using parameters from the DFT calculation[25].

physics in the absence of magnetic field. The interplay between atomic-level relaxation effects and moiré-scale electrostatic potential variation opens new avenues for the design of intertwined topological and correlated states.

## Results

Following recent experiments[6,13–16], we will focus on the valence bands of R-type twisted TMD homobilayers. In these systems, the emergence of nontrivial band topology can be understood as a consequence of the real-space layer pseudospin texture. Within the continuum model, the effective Hamiltonian for the $K$-valley electrons is given by[20]

$$\mathcal{H}_K^\uparrow = \begin{pmatrix} -\frac{\hbar^2(\mathbf{k}-\mathbf{K}_b)^2}{2m^*} + \Delta_b(\mathbf{r}) & \Delta_T(\mathbf{r}) \\ \Delta_T^\dagger(\mathbf{r}) & -\frac{\hbar^2(\mathbf{k}-\mathbf{K}_t)^2}{2m^*} + \Delta_t(\mathbf{r}) \end{pmatrix}, \qquad (1)$$

where $\Delta_{b/t}(\mathbf{r})$ and $\Delta_T(\mathbf{r})$ are the intra- and inter-layer moiré potential, respectively. Due to the spin-valley coupling, the band edge at each valley is spin split, and opposite valleys carry opposite spins as required by time-reversal symmetry[27,28]. The continuum Hamiltonian for the $K'$-valley spin-down electrons can be obtained by applying time-reversal symmetry to $\mathcal{H}_K^\uparrow$, resulting in moiré bands with opposite Chern numbers.

The moiré potential can be represented as an effective layer pseudospin magnetic field $\boldsymbol{\Delta}(\mathbf{r}) = (\,\mathrm{Re}\,\Delta_T, -\,\mathrm{Im}\,\Delta_T, \frac{\Delta_b - \Delta_t}{2})$. There are three high-symmetry local stackings in a moiré supercell, labeled as MM, XM, and MX (Fig. 1). It has been shown that $\boldsymbol{\Delta}(\mathbf{r})$ forms a skyrmion lattice with its north/south poles located at the MX and XM points[20]. Curiously, for 3.89° tMoTe₂, using parameters from the local stacking approximation[20] and the DFT calculation[25], we find a reversal in the positions of the north/south poles between the two cases as shown in Fig. 1, which results in opposite skyrmion numbers[29]. This contrast in skyrmion numbers, in turn, manifests as opposite Chern numbers for the topmost valence band, with only the DFT calculation matching the experiment.

Armed with the insight that the moiré potential landscape can affect the band topology, we now perform DFT calculations at even smaller twist angles. Because the system size at these angles (~13,000 atoms at 1.25°) is beyond the typical scale of DFT relaxations, we first trained a neural network (NN) inter-atomic potential to capture the moiré lattice reconstruction. The NN potentials are parameterized by using the deep potential molecular dynamics (DPMD) method[30,31], where the training data are obtained from 5000 to 6000 ab initio molecular dynamics (AIMD) steps at 500 K for a 6° twisted homobilayer calculated using the VASP package[32]. We test the NN potential for a moiré bilayer at 5° and obtain a root mean square error of force

<0.04 eV/Å. More details can be found in Supplementary Note 1. Figure 2a, b shows the calculated in-plane displacement field of the top-layer W atoms and interlayer distance in tWSe₂ at 3.15° and 1.25°. The displacement field in the bottom layer shows the opposite pattern. It is clear that while the local stacking varies smoothly at 3.15°, at 1.25° large domains of the MX and XM regions form, with domain walls connecting the shrunken MM region. The difference between reconstruction patterns at the large and small twist angles also affects the strain tensor distributions, shown in Supplementary Fig. 6. We find that the shear strain ($u_{xy}$) and $u_{xx} - u_{yy}$ are much larger than the normal strain ($u_{xx} + u_{yy}$). As the twist angle decreases, the strains are mostly distributed near the domain boundaries due to the domain wall formation. These findings are consistent with previous calculations based on continuum model and parameterized inter-atomic potential[26,33–36], as well as available experiments[37–43].

We then calculate the moiré band structure for the relaxed atomic structures. To reduce the computational cost, we adopt the SIESTA package[44] for band structure calculations. We first benchmark the accuracy of this local basis approach with the plane-wave basis approach by comparing the band structures at 6° obtained from SIESTA and VASP, and reach a qualitative agreement between the two (see details in Supplementary Note 4). Then we perform small twist-angle band calculations by using the SIESTA package. Figure 2c, d shows the twist-angle dependence of band structures for tWSe₂ and tMoTe₂, respectively. The top valence bands consist of folded $K$-valley and $K'$-valley minibands with opposite spins. A small band splitting can be seen, mostly between $\gamma$ and $m$. Multiple factors, including trigonal warping and intervalley coupling, may contribute to the splitting. Nevertheless, we assume approximate spin $z$ conservation, and separate moiré bands originating from the two valleys by adding a small Zeeman field in the calculation. In the following, we shall focus on the moiré bands from the spin-up $K$-valley.

To determine the Chern numbers for the moiré bands, we first calculate the eigenvalues of the DFT wave functions under three-fold rotational symmetry ($C_{3z}$). The Chern number is then determined by the product of $C_{3z}$ eigenvalues at rotationally invariant momenta[45]: $\exp(i\frac{2\pi}{3}C) = -\xi_\gamma \xi_\kappa \xi_{\kappa'}$, where the $\xi$'s are the $C_{3z}$ eigenvalue at the high-symmetry point of the moiré Brillouin zone (mBZ). These eigenvalues are labeled in Fig. 2. Our assignment of the Chern numbers for tWSe₂ at 3.15° and 1.70° agree with a recent calculation at 2.28° in which the Chern numbers were calculated by directly integrating the Berry phase over the entire mBZ[26]. The Chern numbers for tMoTe₂ are further confirmed through the integration of the Berry curvature within the mBZ with the help of Wannier interpolations. The change of the Chern numbers with varying twist angles can be understood by tracking the evolution of the $C_{3z}$ eigenvalues, which signals band inversion. For

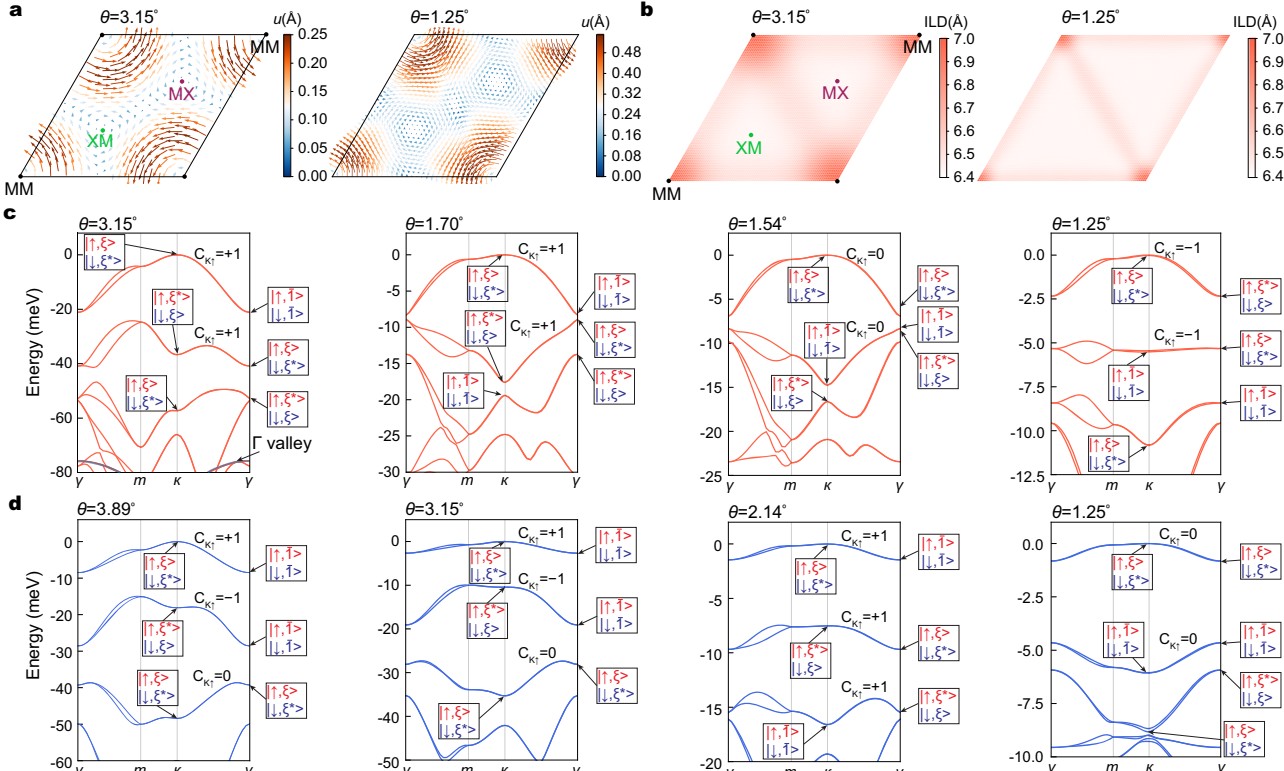

**Fig. 2 | Lattice relaxations and band structures. a** The in-plane displacement field of the top-layer W atoms at 3.15° and 1.25° for tWSe$_2$. The color and arrow denote the amplitude and direction of in-plane displacement fields, respectively. **b** The interlayer distance (ILD) distribution at 3.15° and 1.25° for tWSe$_2$. **c, d** Twist-angle dependence of the valence moiré bands of tWSe$_2$ and tMoTe$_2$, respectively. The labels indicate the spin orientations and the $C_{3z}$ eigenvalues at high-symmetry point with $\xi = e^{i\pi/3}$, $\bar{\xi} = e^{-i\pi/3}$, and $\bar{1} = -1$. The $C_{3z}$ eigenvalue is the same at the $\kappa$ and the $\kappa'$ point.

example, in tWSe$_2$ (Fig. 2c), when the twist angle changes from 1.70° to 1.54°, the first and second bands invert at the $\gamma$ point, and the second and third bands invert at the $\kappa$ and $\kappa'$ points. As a result, the Chern numbers of the two topmost bands change from (+1, +1) to (0, 0).

The twist-angle dependence of the Chern numbers is in excellent agreement with experiments. First, it was reported that at $\nu = -1$ the Chern numbers are opposite for tWSe$_2$ at 1.23°[6] and tMoTe$_2$ at 3.7°[13–16]. Second, in 1.23° tWSe$_2$, the Chern numbers are the same at filling factors $\nu = -1$ and $\nu = -3$, which indicates that the Chern numbers of the first two bands from the same valley must be the same[6]. Further increasing the twist-angle results in trivial insulators up to 1.6°[6]. Remarkably, all these observations are consistent with the trend in the twist-angle dependence of our calculations, confirming the validity of our machine learning-based approach. Our calculation also predicts that in tMoTe$_2$, as the twist angle decreases, the Chern numbers of the two topmost bands change from (+1, −1) to (+1, +1), and finally to (0, 0) at the smallest angle of the calculation. In particular, our calculations reveal multiple flat bands with Chern numbers all equal to +1 at around 2°, indicating the possibility of mimicking higher Landau-level physics in the absence of magnetic field. The presence of multiple bands of Chern number +1 has been confirmed by a recent experimental measurement[46].

Since we are interested in the sign change of the Chern number of the topmost band, in the following we will focus on tWSe$_2$. As mentioned earlier, the evolution of band topology in momentum space is closely related to the change in the real-space moiré potential. In particular, the location of the north/south poles of $\boldsymbol{\Delta}(\mathbf{r})$, which directly affects the skyrmion number, is given by the difference between the moiré potentials at the top and bottom layer.

The moiré potential can be inferred from the surface Hartree potential[47], defined as the difference between the Hartree potential

above and below the twisted bilayer surface in DFT calculations. Figure 3a shows the coarse-grained surface potential drop at 3.15° in tWSe$_2$. More details can be found in Supplementary Note 4. The maximum is located at MX, zero at MM, and minimum at XM. Surprisingly, the surface potential drop shows a sign reversal at XM (and MX) as the twist angles decrease (see Fig. 3a–d). Going from 3.15°, to 1.70°, 1.47°, and eventually down to 1.25°, the potential at the high-symmetry point XM (and MX) changes sign, and the area of the flipped region grows in size. This sign switch suggests that the north pole of $\boldsymbol{\Delta}(\mathbf{r})$ at ~3° becomes the south pole at ~1°. Additional features can be identified near the MM site, where the surface potential drop mimics the pattern of a six-petal flower with $C_3$ symmetry. We find that the amplitude of potential inside the petal is comparable with that at XM (and MX) at 1.70°, suggesting unique quantum confinement effects which reshapes the electron wave function. The overall effects of these features can be clearly seen by a line cut along MM–XM–MX–MM, showing rich variations and multiple extremes in Fig. 3e. The intricate behavior of the surface potential goes beyond the continuum approximation of moiré potential based on the first-star expansion of the reciprocal lattice vectors alone, evident by their Fourier transform as shown in Fig. 3f.

The evolution of the surface potential implies that the layer polarization of the wave functions should also change with the twist angle. In Fig. 4, we plot the real-space wave function for the two topmost bands at the $\gamma$ point of the mBZ at various twist angles for tWSe$_2$. Since the time-reversal symmetry $\mathcal{T}$ and $C_{2x}$ symmetry are preserved at the $\gamma$ point, only the wave function in the top layer is plotted, while the wave function in the bottom layer can be obtained by performing a $\mathcal{T}C_{2x}$ operation, under which MX/XM in the top layer is mapped to XM/MX in the bottom layer. At 3.15°, the wave function of the first band in the top layer is localized at MX. As the twist angle decreases, the

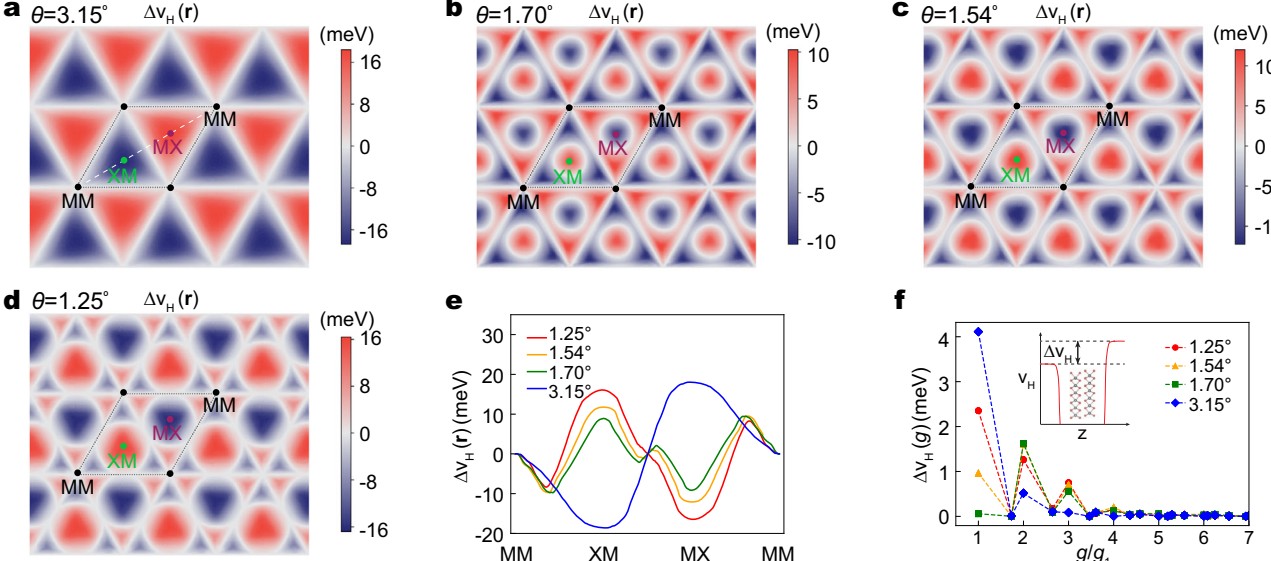

**Fig. 3 | Evolution of surface moiré potential. a–d** Twist-angle dependence of the difference of DFT Hartree potential, $\Delta v_H$, between the top-layer surface and the bottom-layer surface in tWSe₂. The white dashed line denotes the path MM–XM–MX–MM. **e** The distribution of $\Delta v_H$ along the dashed line in (**a**) for different twist angles. **f** The Fourier components of $\Delta v_H$ for different twist angles. $g_1$ is the length of the moiré reciprocal lattice vector. Inset in (**f**): schematics of the Hartree potential drop between the top-layer surface and bottom-layer surface.

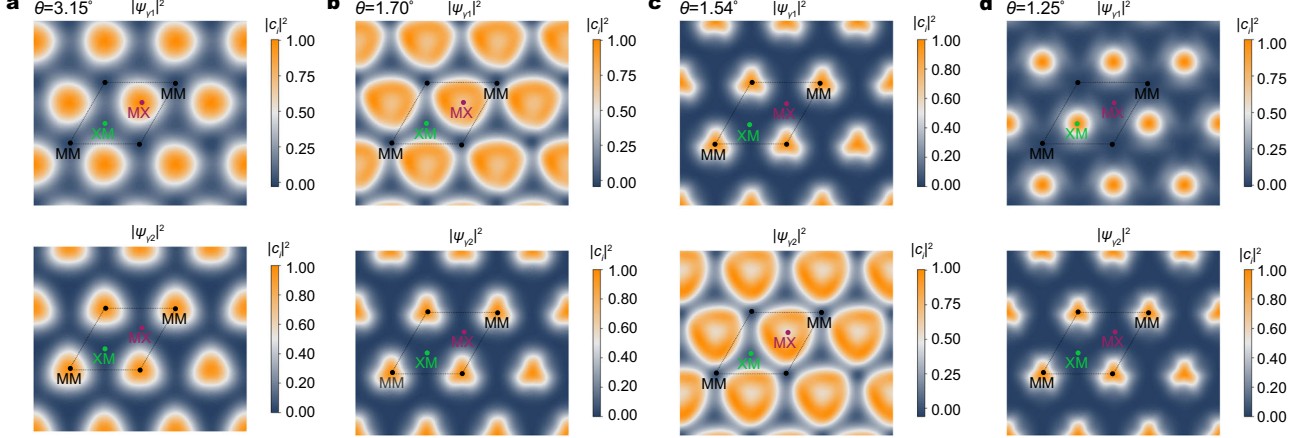

**Fig. 4 | Evolution of wave functions. a–d** Twist-angle dependence of the wave function of the first band (top row) and second band (bottom row) at the $\gamma$ point for the top layer in tWSe₂, respectively. Here, we map the weight of the projected wave function onto the W atomic orbitals. The value is normalized to its maximum in each plot. The dashed parallelogram denotes a moiré unit cell.

localization region switches to MM and eventually to XM. The shift of the wave function location also coincides with the change of the $C_{3z}$ eigenvalue at the $\gamma$ point (Fig. 2c). Similar changes have been found for the wave function of the second moiré band.

Two remarks are in order. First, the switch from MX to XM of the first band indicates the flip of the layer pseudospin, which gives rise to the sign change of the Chern number as shown in Fig. 2c. Second, aside from orbitals located at MX and XM, those at MM also play a significant role in deciding band topology, evident by the distribution of the wave functions (Fig. 4). Thus for a tight-binding model to properly describe the moiré band topology of tWSe₂, one also needs orbitals from the MM site[48]. This goes beyond the real-space skyrmion picture discussed earlier.

What is the origin behind the change in surface potential drop as a function of twist angle? For a two-dimensional (2D) bilayer system in global charge neutrality, the electrostatic surface potential can be directly associated with the interlayer electric polarization. In twisted TMD homobilayers, two microscopic mechanisms contribute to this polarization: ferroelectricity and piezoelectricity. The ferroelectric effects arise from the inversion symmetry breaking in R-type TMD bilayers and have been termed "moiré ferroelectricity"[49,50]. In a moiré supercell, this leads to alternating out-of-plane ferroelectric polarization depending on the local stacking registry, with opposite dipoles in the XM/MX region[50–53]. On the other hand, since monolayer TMDs lack inversion symmetry, the strain field can produce piezoelectric polarization for each layer[54]. Because the two layers have opposite patterns of atomic displacement fields and the same piezoelectric coefficient, the polarization charge distributions are opposite between the two layers, which can produce a vertical potential drop[35,36]. As pointed out in ref. 36, these two types of polarization charges can be opposite in sign, and their competition will determine the potential drop.

Note that additional in-plane polarizations can also arise from the out-of-plane ferroelectricity and local symmetry breaking[55]. While

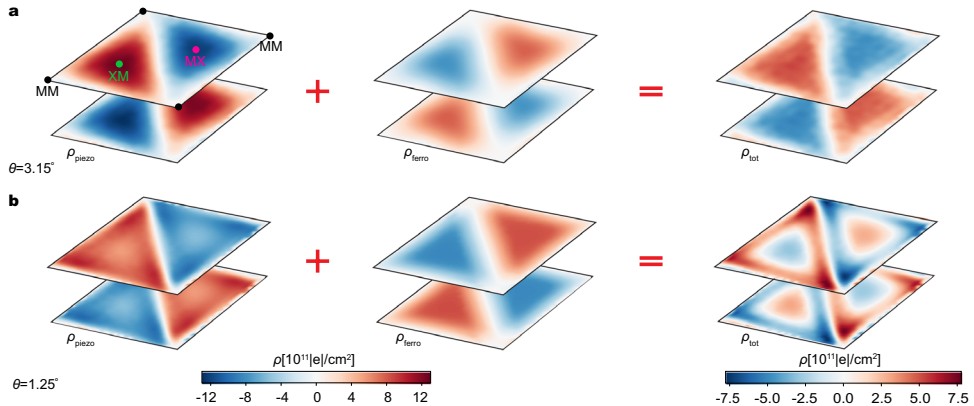

**Fig. 5 | The competition between the piezoelectric and out-of-plane ferroelectric polarizations. a** The piezoelectric charge density, the ferroelectric charge density, and the total charge density, respectively, at 3.15° in tWSe$_2$. **b** The same quantities at 1.25°.

these effects are present in our DFT calculations, they do not create additional Hartree potential that changes the Chern numbers.

Our machine learning-based first-principles simulations enable us to quantitatively study the polarization effects from the relaxed structure with atomic resolution. We start by separately calculating the piezoelectric charge and ferroelectric charge. For each layer, the piezoelectric charge density is directly proportional to the gradient of the strain field by $\rho_{\text{piezo}} = -\tilde{e}_{11}[\partial_x(u_{xx} - u_{yy}) - 2\partial_y u_{xy}]$, where $\tilde{e}_{11}$ is the independent non-zero component of the piezoelectric tensor[35,54]. The out-of-plane ferroelectric polarization is obtained directly from integrating the charge density along the $z$ direction for each local stacking unit cell. More details about the polarization calculations can be found in Supplementary Note 3 and "Methods". In Fig. 5, we plot the piezoelectric (further considering the dielectric screening effect[35]) and ferroelectric charge densities, as well as their sum at 3.15° and 1.25°, respectively. The total polarization charge matches qualitatively the pattern of the surface potential drop.

Specifically, at 3.15°, the piezoelectric charges are mainly distributed in the XM and MX regions because of the larger gradient of shear strain and $u_{xx} - u_{yy}$ in these regions (see Supplementary Fig. 6). The piezoelectric charge is negative at MX and positive at XM for the top layer. Because the bottom layer has the opposite atomic displacements, it has the opposite charge distributions. In contrast, the ferroelectric charge is positive (negative) at MX and negative (positive) at XM for the top (bottom) layer. Adding them together, we find the total charge density is negative (positive) at MX and positive (negative) at XM for the top (bottom) layer. As the twist angle decreases to 1.25°, the ferroelectric charge density at MX and XM remains virtually unchanged, but the total amount of ferroelectric charge within the MX and XM domains increases following the formation of the domain wall. In contrast, because the shear strain and $u_{xx} - u_{yy}$ are mainly distributed along the domain wall and are uniformly small inside the XM and MX domains (see Supplementary Fig. 6), the piezoelectric charge density peaks near the domain wall but decreases at the interior of the domain. This explains the six-petal flower pattern that we discovered for the surface potential drop. As a consequence, the total charge density is now positive (negative) at MX and negative (positive) at XM for the top (bottom) layer. This trend from 3.15° to 1.25° is consistent with the variation of the surface moiré potentials and wave functions. In contrast, within the local stacking approximation, the sign of polarization charge is fixed and a sign reversal of the Chern number is not possible. Probing the predicted reversal of the polarization charges at MX and XM should be a clear experimental evidence of the change of band topology in momentum space.

## Discussion

Up to this point, our discussion has focused on tWSe$_2$. While the evolution of the moiré potential in tMoTe$_2$ follows a similar trend, there are some quantitative differences in the band structures between tWSe$_2$ and tMoTe$_2$. This is mostly due to a couple of factors. WSe$_2$ has a lighter effective mass (0.35) compared to MoTe$_2$ (0.62), resulting in a larger bandwidth. In addition, differences in both elastic and piezoelectric coefficients also lead to quantitative changes in the moiré potential between these materials. More details can be found in Supplementary Information.

In summary, we have performed large-scale DFT calculations on R-type TMD homobilayers. Our results demonstrate machine learning as a powerful tool to study moiré systems, by revealing the importance of lattice relaxation that eventually leads to qualitative changes in band topology. This change is attributed to the competition between piezoelectricity and out-of-plane ferroelectricity, resulting in electrostatic potential variation that reshapes the potential landscape for any moiré electronic states. Our findings highlight the crucial long-range interactions arising from polarization charges, which change rapidly as the twist angle decreases. This behavior necessitates the explicit calculations of moiré electronic potential even at minimal twist angle.

Note added. We recently became aware of two related works in which machine learning force field is also used to calculate the moiré bands of twisted MoTe$_2$ homobilayers[56,57].

## Methods
### Machine learning
AIMD simulations are conducted to generate training datasets. These simulations utilize the VASP package[32], employing the projector augmented wave pseudopotential[58,59] and the Perdew–Burke–Ernzerhof (PBE) exchange-correlation functional[60]. Additionally, van der Waals corrections are incorporated using the D2 formalism[61]. 5000-step AIMD simulations using the canonical ensemble are performed at 500 K for 6° tMoTe$_2$, and 6000-step for 6° tWSe$_2$. The NN inter-atomic potential is generated using the DPMD method[30,31]. One thousand steps of the training data are used for validations. The embedding and fitting NNs include three hidden layers and the cutoff radius for each atom is 10.0 Å. One million steps (batches) with a batch size of 1 are used to minimize the loss function that includes energy and force contributions. One hundred new steps of MD trajectories at 500 K are used to test the NN potentials. The loss functions and comparisons between the NN inferences and DFT calculations can be found in Supplementary Note 1. The NN potentials are used to relax the superlattice within the LAMMPS package[62] until the maximum atomic force is smaller than $10^{-4}$ eV/Å.

## Band structure calculations

The SIESTA package is used to calculate band structures. Optimized norm-conserving Vanderbilt pseudopotentials[63], PBE functional[60], and double-zeta plus polarization basis are used. Spin–orbit coupling (SOC) is treated within the on-site approach[64]. We first perform self-consistent calculations without SOC. Then we include on-site SOC without iterating charge densities. The validations of the on-site SOC and the comparisons between the band structures from SIESTA and VASP can be found in Supplementary Note 4.

## Electric polarization calculations

The proper piezoelectric coefficient is defined as[65]

$$\tilde{e}_{ijk} = \left( \frac{\partial J_i}{\partial \dot{u}_{jk}} \right)_{E,T}, \tag{2}$$

which represents the response of the current with respect to the strain flow. Here $J_i$ is the current component, $u_{jk}$ the strain component, $E$ the macroscopic electric field, and $T$ is the stress. Both $E$ and $T$ are zero in the DFT calculations. Since monolayer $WSe_2$ has the symmetry of $D_{3h}$, the only independent non-zero piezoelectric coefficient is $\tilde{e}_{11} \equiv \tilde{e}_{111}$ and other non-zero coefficients are related to $\tilde{e}_{11}$ by[54]

$$\tilde{e}_{122} = -\tilde{e}_{11}, \tag{3}$$

$$\tilde{e}_{212} = \tilde{e}_{221} = -\tilde{e}_{11}. \tag{4}$$

$\tilde{e}_{11}$ is calculated from the change of electric polarization density with respect to the strain[54]. A rectangular unit cell of monolayer $WSe_2$ and a $12 \times 12 \times 1$ $k$-space grid are used. The in-plane polarization is calculated in SIESTA using the modern theory of polarization[66].

The piezoelectric polarizations in the moiré superlattice are calculated from the strain,

$$\mathbf{P} = \tilde{e}_{11}(u_{xx} - u_{yy}, -2u_{xy}), \tag{5}$$

and the piezoelectric charge density is calculated as

$$\rho_{piezo} = -\nabla \cdot \mathbf{P} = -\tilde{e}_{11}[\partial_x(u_{xx} - u_{yy}) - 2\partial_y u_{xy}]. \tag{6}$$

More details can be found in Supplementary Note 3. We use the method in ref. 35 to include the dielectric screening of piezoelectric charges.

The out-of-plane moiré ferroelectricity is calculated from the local stacking unit cell in the relaxed moiré superlattice. Within each unit cell, we calculate the out-of-plane dipole moment in SIESTA by integrating the charge density multiplied by $z$ coordinates. Then the surface charge density due to ferroelectricity is obtained as $\rho_{ferro} = P_z/(Sd_z)$, where $P_z$ is the dipole moment, $S$ is the area of the unit cell, and $d_z$ is the interlayer vertical distance between transition metal atoms.

## Data availability

The files and datasets generated during this study, including typical input files for parameterizing NN potentials and DFT calculations, as well as the source data of band structures, Hartree potentials, wave functions, and polarizations, have been deposited in the Zenodo database[67]. Other data related to this paper are available from the corresponding authors upon request.

## Code availability

The DFT calculations and parameterization of NN potentials are obtained from publicly available packages, following the procedure outlined in the paper. Automation and acceleration workflows are available from the corresponding authors on request.

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

## Acknowledgements

We acknowledge useful discussions with Jiaqi Cai, Ken Shih, and Xiaodong Xu. X.W.Z. is supported by DOE Award No. DE-SC0012509. C.W. is supported by ARO MURI with award number W911NF-21-1-0327. Y.F. is supported by the discovering AI@UW Initiative and DMR-2308979. This work was facilitated through the use of advanced computational, storage, and networking infrastructure provided by the Hyak supercomputer system and funded by the University of Washington Molecular Engineering Materials Center at the University of Washington (DMR-2308979). Access to GPUs is provided by Microsoft's sponsorship of Azure credits to the Department of Materials Science and Engineering at UW.

## Author contributions

X.W.Z., T.C., and D.X. conceived the project. X.W.Z. performed the parameterization of NN potentials, DFT calculations, and polarization calculations with the help of C.W., T.C., and D.X. X.W.Z., T.C., and D.X. analyzed the results and wrote the paper with input from C.W., X.L., and Y.F.

## Competing interests

The authors declare no competing interests.
