## [Peer Review File · Nature Communications]

Polarization-driven band topology evolution in twisted MoTe₂ and WSe₂Reviewers' Comments:

Reviewer #1:

Remarks to the Author:

The authors study the band topology and electrostatic properties of twisted MoTe₂ and WSe₂ bilayers. They perform large-scale first-principles calculations to obtain electronic properties, using machine learning force fields to obtain relaxed structures.

While the calculations are impressive and thorough, I do not think there is enough novelty to warrant publication in Nature Communications, for the reasons listed below.

More importantly, I am not convinced by the central argument of the paper, that the changes in band topology with twist angle are driven by the interplay between the different types of polarization.

- The central claim is that the changes in the Chern numbers shown in Fig. 2 are driven by a change in the electrostatic potential in Fig. 3. Going from 3.15 to 1.70, there is a change in the shape of the Hartree potential. Small bubbles appear inside the triangular domains, which the authors claim arises due to piezoelectricity. However, below this angle, there are additional changes in the Chern numbers, while the general shape of the potential remains the same, with the bubbles becoming bigger. I am not convinced that such a dramatic change in band topology could arise from a small and continuous change in the electrostatic potential.

- This appearance of bubbles inside the stacking domains in the electrostatic potential should be a second-order (continuous) effect. Perhaps it would be constructive to locate the point where this occurs as a function of twist angle. If the band topology really only changes before and after this point, then perhaps that would be more convincing evidence. Also, doing larger twist angles where the electrostatic potential is completely triangular and showing that the band topology stays the same might help too.

- Is the Hartree potential in Fig. 3 really a direct output of siesta? The authors should provide more details on how it was calculated. From the description in the paper, it seems like it is the potential drop across the bilayer, not averaged in-plane, is this correct? Has some in-plane averaging been done to smooth out the atomic-scale oscillations?

- If this really is a direct output from a DFT calculation, then the change in the electrostatic potential with twist angle is indeed very interesting. But I am not convinced this is coming from a balance between contributions to the polarization. How can the in-plane polarization arising from piezoelectricity affect a vertical potential drop? Additionally, as the twist angle decreases, the strain inside the domain centers becomes minimized, with most of the strain confined to the domain walls and around the AA stacking points. This is well known from many other theoretical studies, and the authors even show it in Fig. 2. How then, can the piezoelectric polarization become larger with decreasing twist angle, when the strain inside the domains is decreasing?

- While I think the use of a MLFF is nice and may be useful for the community, I don't think this adds to the novelty of the paper. The structural properties of twisted bilayers are already very well-understood, from continuum elastic field theories, classical MD, and even full DFT relaxations. In addition, there are several other MLFFs for the same materials available, some of which the authors cite. Can we learn anything new about the structural properties of twisted bilayers using MLFFs that can't be found using the aforementioned methods? This is not a criticism of the work, but a comment on the novelty of the method.

A few additional comments:

- The authors refer to the out-of-plane polarization as "ferroelectric polarization" throughout the paper. This is not accurate. The polar domains (AB/BA) arising from interlayer transfer between the layers (due to symmetry breaking of different stackings) cannot be inverted at zero field, and are therefore not ferroelectric. I would recommend referring to them simply as polar domains. In an untwisted bilayer the polarization can be reversibly switched at zero field via van der Waals sliding, but this is not the case in twisted bilayers. A subtle point, but I think it is important to be accurate here.

- I don't like the treatment of polarization in terms of charges, i.e. "ferroelectric charge" and "piezoelectric charge". All of the information about the direction of the polarization is lost, and it is an

arbitrary decomposition of the charge. The authors cite works which have also done this, but I think the decomposition of the charge into terms arising from different components of the polarization is somewhat arbitrary and misleading. For example, the polarization arising from piezoelectricity is in-plane, but this is not evident from the paper, in particular Fig. 5. This is then compared to the vertical potential drop in Fig. 3, which I think is misleading.

- I am not sure that the treatment of piezoelectricity here (or in the references cited) is correct/complete. The authors use the piezoelectric tensor for a monolayer, and then use the strain field derived from the twisted bilayer to obtain the piezoelectric contribution to the polarization. However, the piezoelectric tensor for a twisted bilayer may differ significantly, since the symmetry properties of the bilayer are different. In particular, the symmetries of the different stackings (AB, BA, AA, domain wall) may mean some additional components are nonzero in different domains, and may contribute additional terms. To my knowledge, a detailed study of this has not been done, but I think the treatment here is too simplistic. Finally, the authors should be careful to ensure that they calculate the "proper" piezoelectric response (see e.g. arXiv:cond-mat/9903137), i.e. measure the difference in potential drop with respect to strain, rather than electric field.

- Additionally, even neglecting the piezoelectric polarization, twisted bilayers already have an in-plane polarization arising from the same symmetry breaking which gives rise to the out-of-plane polarization. This is not discussed here, and while I do not think this should affect the Hartree potential either, should be considered alongside the piezoelectric polarization.

And some minor comments:

- The authors develop a MLFF using vasp, then switch to siesta for the DFT calculation. What is the point in training a MLFF in one code, then using it with another code? Why not just train the MLFF using siesta?

- I don't understand why the pseudospin skyrmion lattice is mentioned at all here (Fig. 1 and throughout the text). To my knowledge, the pseudospin is a particular interpretation of the Chern number as a winding number (Pontryagin), only for effective Hamiltonians which can be cast in a $k \cdot \sigma$ form. Why bother referring to this in a paper about full DFT calculations, which does not contain any effective models?

- Additionally, the authors use the terms Chern number and skyrmion interchangeably throughout the paper, which would be confusing for a non-specialist.

- Fig. 2 a is very difficult to read. Consider plotting the displacement and layer separation separately.

- In Fig. 3 f, the Fourier components of the Hartree potential all seem to become negligible beyond 3. I would have expected them to decay more slowly for decreasing twist angle, when relaxation becomes more significant.

- In Fig. 4 c, lower panel, it looks like C3 symmetry is broken, but this is not the case in any of the others. What is going on here?

Reviewer #2:

Remarks to the Author:

Review Report on manuscript NCOMMS-23-57855, "Polarization-driven band topology evolution in twisted MoTe2 and WSe2"

The manuscript presents multi-scale calculations to accurately investigate the moiré systems from large to small twist angles. Large-scale density-functional-theory (DFT) calculations with machine learning force fields are used for the structure relaxation and moiré band topology. Such systems are attracting significant attention. The authors found that the Chern numbers of the moiré frontier bands change sign as a function of twist angle due to the competition between the in-plane piezoelectricity and the out-of-plane ferroelectricity. This work not only provided a reliable explanation of recent experimental observations in twisted MoTe2 and WSe2 homobilayers. Since the tunable property of a

moiré system is sensitive to structural details, an accurate and reliable analysis here is essential. The analysis method could also be extended to other twisted systems. The results of the manuscript are devoted to a hot research topic and are of interest in the field of condensed matter physics. The manuscript can be therefore considered for publication in Nature Communications.

I also have some comments or suggestions:

1. As the authors emphasized, the pattern of surface potential changes when the angle is smaller than 1.7° in $tWSe_2$ systems, where the in-plane piezoelectric is relatively enhanced. I am curious on the local structure where the in-plane piezoelectric is large. Whether the local structure is still R-type or H-type stacking? It is recommended that the amplitude and unit of the arrow, namely in-plane displacement field, in Fig. 2(a), and the local structures with maximal displacement should be given.

2. The authors generated the training data sets in R-type 6° twisted systems and tested them in 5° systems. Compared to DFT results, the authors claimed the validity of the machine learning model. However, I am concerned about this validity in systems with even smaller twisted angles. If the local stacking is H-type, the systematical comparison would be better via training the model in H-type 6° (namely R-type 54°) twisted data sets.

3. The method details on the surface moiré potential should be given. The amplitude of surface moiré potential presented in this work is somehow smaller than that reported by Jiang Zeng et al. in reference PRB 107, L081402 (2023). In this reference, the authors also reported that the out-plane polarization might not dominate, which coincides with the result of the present work.

Typo: "angels" in the abstract should be corrected to "angles"

Reviewer #3:

Remarks to the Author:

The authors report on large-scale DFT calculations for twisted bilayers of WSe_2 and $MoTe_2$ at small twist angles, considering lattice relaxation using molecular dynamics with potentials trained using machine learning plus DFT.

The central result of this paper is as follows: a careful treatment of lattice relaxation provides a more accurate moiré potential, which allowed the authors to reproduce fine experimental results, particularly the Chern numbers of the topologically trivial or non-trivial top valence moiré minibands, and their dependence on twist angle. The authors provide a theoretical explanation to recent experimental results that have established different Chern numbers for the top valence minibands of twisted WSe_2 and $MoTe_2$; a fact that was not properly explained by continuous models thus far. The authors argue that the different topological invariants for the two materials originate from an inversion of the interlayer electrostatic potential drop at XM and MX domains in the reconstructed moiré superlattice for twisted WSe_2 , for twist angles around 1.4° , and which is absent for $MoTe_2$, at least down to 1.25° according to the authors' calculations.

In my opinion, this paper is both sound and timely, in addition to having direct experimental impact. I therefore recommend its publication in Nat. Commun., once the questions and comments below are properly addressed by the authors:

1. As I mentioned above, the central result of this paper is that lattice reconstruction must be carefully considered in order to obtain an accurate moiré potential. Here, the authors achieve this by relaxing the structures using machine-learning-trained molecular dynamics potentials plus DFT calculations. However, there is a significant body of work by the Manchester theory group, including Refs. [36,37,40,53] of the present manuscript where lattice relaxation is estimated using a different

approach (DFT + elasticity theory). It is not clear how and if the authors' approach differs or improves upon the results obtained using the latter mentioned techniques. Can the authors comment on that? In particular, Ref. [53] deals with bilayer WSe₂, which should provide with a suitable point of comparison

2. I believe the work by Pan et al. on the band topology of twisted WSe₂ bilayers from 2020 merits citation: [Phys. Rev. Research 2, 033087 (2020)]. This should also nicely complement the comparisons with Ref. [20], which studied the case of MoTe₂. Both references use continuous models limited to the first star of moiré Bragg vectors (neglect reconstruction)

3. In line 127 of the manuscript, the authors mention a computational-cost-saving measure which I, as a non DFT specialist, did not understand even after reading Supplementary Sec. IV. The authors mention using both SIESTA and VASP, which sounds like double the work. It seems to me that one package deals with SOI while the other doesn't, but I would prefer it if the authors themselves could clarify this point, especially in the main text.

4. Likewise, the comment at line 140 about introducing an artificial Zeeman splitting between the two valleys is not entirely clear to me. Was the purpose of this to minimize intervalley mixing by the moiré potential? If so, how large was the Zeeman splitting introduced, and how does it compare with intervalley mixing terms? I could not find this discussion in the Supplementary Materials.

5. Finally, in line 212 the authors state that "The intricate behavior of the surface potential goes beyond the continuum approximation of moiré potential based on the first-star expansion of the reciprocal lattice vectors alone (...)", with which I agree. Although Ref. [20] and my strongly suggested reference [Phys. Rev. Research 2, 033087 (2020)] do stop at first stars of moiré vectors, Ref. [53] of the present manuscript does formulate multiple continuous models (for R and H cases and for different points in the Brillouin zone) that incorporate lattice reconstruction, and which include multiple stars of Bragg vectors. Indeed, Fig. 16b of that reference matches the spatial distribution of holes reported by the authors in Fig. 4c for about the same angle. Can the authors comment on how those results compare with theirs, and on whether they believe that, once the reconstruction has been properly calculated, the results of the present manuscript could be reproduced by a continuous model that incorporates that reconstruction? I believe the authors' input on this matter will be valuable to readers.

Response to Reviewers' comments

We greatly appreciate the reviewers' feedback. Below, we respond to their comments in a point-by-point format. All major changes are highlighted in red font in the revised manuscript.

Report of Review #1

Comment 1: The authors study the band topology and electrostatic properties of twisted MoTe₂ and WSe₂ bilayers. They perform large-scale first-principles calculations to obtain electronic properties, using machine learning force fields to obtain relaxed structures. While the calculations are impressive and thorough, I do not think there is enough novelty to warrant publication in Nature Communications, for the reasons listed below. More importantly, I am not convinced by the central argument of the paper, that the changes in band topology with twist angle are driven by the interplay between the different types of polarization.

Our Response: We appreciate the reviewer's praise for our "impressive and thorough calculations" and for their insightful comments. However, we respectfully disagree with their judgment regarding the novelty of our results, due to the facts that the new physical mechanism we discovered not only resolves the puzzles of opposite Chern numbers observed in moiré homobilayers at an integer filling factor, but also predicts new topological band series which serves as a foundation to understand latest discoveries at higher filling factors. Please allow us to explain in more details below.

Our work is motivated by the recent experimental discovery of fractional Chern insulators (FCI) in twisted transition metal dichalcogenide (TMD) homobilayers, which has sparked significant interest in the condensed matter physics and materials science communities. Unlike the quantum Hall effect in a strong magnetic field, which exhibits Landau levels, the FCI is characterized by flat Chern bands, offering far greater tuning capabilities. Understanding the formation of these flat Chern bands at the single-particle level is therefore of crucial importance. Despite the initial excitement, several experimental puzzles have arisen. One of the most perplexing is the observation of opposite Chern numbers in 1.25° twisted WSe₂ (tWSe₂) and 3.89° twisted MoTe₂ (tMoTe₂). As pointed out by reviewer 3, current theories based on the continuum model cannot explain this difference, which highlight the need for a comprehensive understanding of band topology in twisted TMD homobilayers. Using machine learning assisted large-scale DFT calculations, we have not only addressed this discrepancy but have also made predictions of the Chern numbers over a wide range of twist angles. In fact, after our paper was posted online, the Chern numbers of the topmost three bands have also been measured in 2.1° tMoTe₂ [K. Kang, et al, Evidence of the fractional quantum spin Hall effect in moiré MoTe₂, Nature (2024)], which strongly endorses our predictions.

The information of the band topology is not accessible using conventional DFT calculations, and the swift impact of our work on moiré research, as well as its potential to guide further experimental investigations in fractional Chern insulator, justifies the novelty and importance of our findings.

Comment 2: The central claim is that the changes in the Chern numbers shown in Fig. 2 are driven by a change in the electrostatic potential in Fig. 3. Going from 3.15 to 1.70, there is a change in the shape of the Hartree potential. Small bubbles appear inside the triangular domains, which the authors claim arises due to piezoelectricity. However, below this angle, there are additional changes in the Chern numbers, while the general shape of the potential remains the same, with the bubbles becoming bigger. I am not convinced that such a dramatic change in band topology could arise from a small and continuous change in the

electrostatic potential. This appearance of bubbles inside the stacking domains in the electrostatic potential should be a second-order (continuous) effect. Perhaps it would be constructive to locate the point where this occurs as a function of twist angle. If the band topology really only changes before and after this point, then perhaps that would be more convincing evidence. Also, doing larger twist angles where the electrostatic potential is completely triangular and showing that the band topology stays the same might help too.

Our Response: We thank the reviewer for the comment. The critical twist angles of band crossing cannot be sampled in the DFT calculations, due to the constraint of constructing a finite-size moiré supercells. However, we are able to sample twist angles with smaller steps, so that an approximate gap closing behavior can be reached. The result is included in Fig. R1, which clearly shows that between 1.70 and 1.54 degree, band inversions take place at the γ point between the first and second bands and the κ (κ') point between the second and third bands.

Fig.R1. The band structures of tWSe2 at 2.14 degree, (a), 1.70 degree, (b), and 1.54 degree, (c). The labels indicate the spin orientations and the C_{3z} eigenvalues at high symmetry points with $\xi = e^{i\pi/3}$, $\xi^* = e^{-i\pi/3}$ and $\bar{1} = -1$.

The physical mechanism of this change of topology is that the electrostatic potential variation is significant, rather than “small” or “second-order” as the referee suggested, due to the following reasons.

(1) The potential variation is actually comparable to the band gap and band width. For example, between 1.70 to 1.25 degree, the inter-layer potential drop changes by ~ 8 meV at MX and XM sites. The bandwidth of the first band is smaller than 10 meV, and the band gap between the first and second bands is smaller than 5 meV.

(2) We have shown in the paper that the wave functions are located at the high symmetry points (MX, XM and MM) in the moiré Brillouin zone. Depending on their location, the eigenvalue of the corresponding wave functions under C_3 rotation will be different, which leads to different Chern numbers. For the first moiré band of small kinetic energy, such location is mostly affected by the maxima of the potential. Therefore, it is the position of electrostatic potential maxima rather than its shape (bubbles or not) that determines the Chern number. While the change of the potential is continuous, the change of the location of the potential maxima is not. For example, while the bubble appears between 1.70 and 3.15 degree, the Chern number remain unchanged because the maxima does not move. In contrast, although the bubble within the domain wall gradually expands with decreasing twist angle, the location of potential maxima switches between MM, MX and XM, as depicted in Fig.3(e) (replotted in Fig.R2), and the Chern number

changes. This is also reflected in our plot of the wave function in Fig. 4, where the location of the wave function shows similar switching behavior.

We believe the reasoning provided above has validated our main claim that the changes in the Chern numbers shown in Fig. 2 are driven by a change in the electrostatic potential.

Fig.R2. The distribution of interlayer Hartree potential difference along the path MM-XM-MX-MM at different twist angles.

As suggested by the referee, we include an additional band structure calculated of tWSe2 at 5.08 degree. Due to increasing kinetic energy (larger Brillouin zone), the band becomes more dispersive but the Chern number of the first moiré bands stays +1 in the calculation (Fig. R3). However, we believe the discussion of this regime is outside the scope of our current work that focus on twist angles below 4 degree.

Fig.R3. The band structures of tWSe2 at 5.08 degree, (a), and 3.89 degree, (b).

Comment 3: Is the Hartree potential in Fig. 3 really a direct output of siesta? The authors should provide more details on how it was calculated. From the description in the paper, it seems like it is the potential drop across the bilayer, not averaged in-plane, is this correct? Has some in-plane averaging been done to smooth out the atomic-scale oscillations?

Our response: We thank the reviewer for asking this technical question. The Hartree potential in Fig. 3 is obtained from DFT calculations in SIESTA with post processing. To help with the analysis of moiré potential, we first obtained 2D maps of Hartree potential at ~ 2.5 angstrom above the top layer and below the bottom layer. Then we employ coarse graining techniques to smooth out the oscillations due to intra unit cell atomic structures. This allows us to compare the atomic level calculations with macroscopically defined electric polarization and potential. For example, Fig.R4 (a) and (b) show the original Hartree potential obtained from SIESTA at ~ 2.5 angstrom above the top layer and below the bottom layer for 3.15-degree tWSe₂, respectively. Fig.R4 (c) and (d) show the corresponding smoothed Hartree potential distributions. The difference between the two gives the potential drop reported in the manuscript. We have added the calculation details in the revised Supplementary Materials.

Fig.R4. The Hartree potential directly obtained from SIESTA at ~ 2.5 angstrom above the top layer, (a), and below the bottom layer (b) for 3.15-degree tWSe₂. (c) and (d) resemble (a) and (b) but with coarse graining to smooth out the atomic-scale oscillations.

Comment 4: If this really is a direct output from a DFT calculation, then the change in the electrostatic potential with twist angle is indeed very interesting. But I am not convinced this is coming from a balance between contributions to the polarization. How can the in-plane polarization arising from piezoelectricity affect a vertical potential drop?

Our response: We want to remind the reviewer that we are considering a bilayer system in which the in-plane atomic displacements are opposite in the top and bottom layers. As a result, the in-plane piezoelectric charges in each layer have opposite sign at the same in-plane location (see Fig. 5 and Fig.R6), resulting in an out-of-plane dipole and a vertical potential drop. In addition to the reviewer, we have received similar feedback at workshops and conferences on the confusion between in-plane vs. out-of-plane potential/polarization. To be clear, in the revised manuscript, we change the notation of “in-plane piezoelectric polarization” to “piezoelectric polarization”, and have added the following sentences on page 5:

“On the other hand, since monolayer TMDs lack inversion symmetry, the strain field can produce piezoelectric polarization for each layer~\cite{duerloo2012intrinsic}. Because the two layers have opposite patterns of atomic displacement fields and the same piezoelectric coefficient, the polarization charge distributions are opposite between the two layers, which can produce a vertical potential drop~\cite{enaldiev2020stacking,magorrian2021multifaceted}.”

Comment 5: Additionally, as the twist angle decreases, the strain inside the domain centers becomes minimized, with most of the strain confined to the domain walls and around the AA stacking points. This is well known from many other theoretical studies, and the authors even show it in Fig. 2. How then, can the piezoelectric polarization become larger with decreasing twist angle, when the strain inside the domains is decreasing?

Our response: We respectfully point out that it is a misread of our manuscript that the piezoelectric polarization becomes larger with decreasing twist angle. In fact, we clearly shown in Fig. 5 that the piezoelectric charge density (arising from polarization) becomes smaller inside the domains (MX and MX) at 1.25° than at 3.15°. We explain in detail below.

The piezoelectric charge density for each layer is related to the divergence of the piezoelectric polarization, or the gradient of the strain field by,

$$\rho_{piezo} = -e_{11}[\partial_x(u_{xx} - u_{yy}) - 2\partial_y u_{xy}].$$

Figures R5(a)-(c) show the normal strain ($u_{xx} + u_{yy}$), the shear strain (u_{xy}), and $u_{xx} - u_{yy}$ for the top layer of 3.15-degree tWSe2. It can be seen that although the normal strain is negligible, the shear strain and $u_{xx} - u_{yy}$ are significant and contribute to the piezoelectric polarizations. Because the shear strain and $u_{xx} - u_{yy}$ change quickly around MX and XM sites, the piezoelectric polarization charges are mainly located around MX and XM sites. Figures R5(d)-(f) show the normal strain ($u_{xx} + u_{yy}$), the shear strain (u_{xy}), and $u_{xx} - u_{yy}$ for the top layer of 1.25-degree tWSe2. As the twist angle decreases to 1.25 degree, it can be seen that the shear strain and $u_{xx} - u_{yy}$ are mainly distributed along the domain wall and they become uniformly small at MX and XM. As a result, the piezoelectric charges are pushed into the domain wall and around the MM site. Then the piezoelectric charge density becomes smaller inside the domains (MX and XM) at 1.25 degree than at 3.15 degree. We have added Fig.R5 in the revised Supplemental Materials and modified the following paragraph in the revised manuscript on page 5:

“Specifically, at 3.15°\circ\$, the piezoelectric charges are mainly distributed in the XM and MX regions because of the larger gradient of shear strain and $u_{xx} - u_{yy}$ in these regions (see Fig.S5 of Ref.~\cite{supp}). The piezoelectric charge is negative at MX and positive at XM for the top layer. Because the bottom layer has the opposite atomic displacements, it has the opposite charge distributions. In contrast, the ferroelectric charge is positive (negative) at MX and negative (positive) at XM for the top (bottom) layer. Adding them together, we find the total charge density is negative (positive) at MX and positive (negative) at XM for the top (bottom) layer. As the twist angle decreases to 1.25°\circ\$, the ferroelectric charge density at MX and XM remains virtually unchanged, but the total amount of ferroelectric charge within the MX and XM domains increases following the formation of the domain wall. In contrast, because the shear strain and $u_{xx} - u_{yy}$ are mainly distributed along the domain wall and are uniformly small inside the XM and MX domains (see Fig.S5 of Ref.~\cite{supp}), the piezoelectric charge density peaks near the domain wall but decreases at the interior of the domain. This explains the six-petal flower pattern that we discovered for the surface potential drop. As a consequence, the total charge density is now positive (negative) at MX and negative (positive) at XM for the top (bottom) layer.”

Fig.R5. (a)-(c), the normal strain, the shear strain, and $u_{xx} - u_{yy}$ of the top layer in 3.15 degree tWSe₂, respectively. (d)-(f) are similar to (a)-(c) but show the corresponding strains at 1.25 degree.

Comment 6: While I think the use of a MLFF is nice and may be useful for the community, I don't think this adds to the novelty of the paper. The structural properties of twisted bilayers are already very well-understood, from continuum elastic field theories, classical MD, and even full DFT relaxations. In addition, there are several other MLFFs for the same materials available, some of which the authors cite. Can we learn anything new about the structural properties of twisted bilayers using MLFFs that can't be found using the aforementioned methods? This is not a criticism of the work, but a comment on the novelty of the method.

Our response: While it is true that elastic theories and classical MD can be used to study lattice relaxation for small twist angles, we have to point out that full DFT relaxation is only possible for large twist angles. Even at 4 degree, full relaxation can take 2 weeks. At even smaller twist angles such as 2 degree, full DFT relaxation is almost impossible due to the scaling of computational complexity and numerous local minimum in the Born-Oppenheimer surface.

We also want to point out that the other MLFF paper we cited appeared only weeks earlier than us on arXiv. As a result, we believe our application of MLFF to moiré systems is both new and necessary at the first-principles accuracy, which is a prerequisite to understand the polarization-driven evolution of topology. The methodology that we laid out in this paper can be generalized to more complex moiré superlattices and heterostructures, whereas classical MD approach requires new parameterization.

Comment 7: The authors refer to the out-of-plane polarization as "ferroelectric polarization" throughout the paper. This is not accurate. The polar domains (AB/BA) arising from interlayer transfer between the layers (due to symmetry breaking of different stackings) cannot be inverted at zero field, and are therefore not ferroelectric. I would recommend referring to them simply as polar domains. In an untwisted bilayer the polarization can be reversibly switched at zero field via van der Waals sliding, but this is not the case in twisted bilayers. A subtle point, but I think it is important to be accurate here.

Our response: We agree with the reviewer that in the strictest sense that the polar domains due to interlayer charge transfer is not ferroelectric. However, our naming convention follows "moiré ferroelectricity" which

has already been used in published works (See, for example, Science 372, 1458-1462 (2021)), and is familiar by our potential readers. We chose not to use polar domains because we are discussing the competition between charges associated with the piezoelectricity and the ferroelectricity. We want to make a distinction between these two.

Comment 8: I don't like the treatment of of polarization in terms of charges, i.e. "ferroelectric charge" and "piezoelectric charge". All of the information about the direction of the polarization is lost, and it is an arbitrary decomposition of the charge. The authors cite works which have also done this, but I think the decomposition of the charge into terms arising from different components of the polarization is somewhat arbitrary and misleading. For example, the polarization arising from piezoelectricity is in-plane, but this is not evident from the paper, in particular Fig. 5. This is then compared to the vertical potential drop in Fig.3, which I think is misleading.

Fig.R6. Schematics of the atomic displacement (black curve), piezoelectric charge, and direction of vertical potential drop (red arrow).

Our response: We believe that this comment stems from the same question in Comment 4. In our work, we focus on charge instead of polarization because the charge is a more fundamental quantity than the polarization. Reporting the charge, i.e., the divergence of the total polarization, makes the physics picture more accessible without drawing and adding arrows in 3D. This choice does not, however, affect the analysis or conclusions of our work.

To make the picture more self-explanatory, we schematically plot directions of the vertical potential drop arising from piezoelectric polarization as shown in Fig. R6. Here the piezoelectric charge can be defined by the piezoelectric polarization of each individual layer. The two layers have opposite atomic displacement field, indicated by the black curve with arrows. As a consequence, the polarization charge is opposite in the two layers at the same in-plane coordinates. Fig. R6 shows that the vertical potential drop and charge indeed have one-to-one correspondence. In a capacitor model, the polarization charge and interlayer potential drop are related by the interlayer distance. Therefore, none of these quantities, i.e., polarization, charge, and potential drop, have ambiguity. As we can see now, the direction of the in-plane polarization is not the most straightforward information to understand the vertical potential drop. But the (opposite) charge density in the two layers is.

Comment 9: I am not sure that the treatment of piezoelectricity here (or in the references cited) is correct/complete. The authors use the piezoelectric tensor for a monolayer, and then use the strain field derived from the twisted bilayer to obtain the piezoelectric contribution to the polarization. However, the

piezoelectric tensor for a twisted bilayer may differ significantly, since the symmetry properties of the bilayer are different. In particular, the symmetries of the different stackings (AB, BA, AA, domain wall) may mean some additional components are nonzero in different domains, and may contribute additional terms. To my knowledge, a detailed study of this has not been done, but I think the treatment here is too simplistic. Finally, the authors should be careful to ensure that they calculate the "proper" piezoelectric response (see e.g. arXiv:cond-mat/9903137), i.e. measure the difference in potential drop with respect to strain, rather than electric field.

Our response: We respectfully disagree with the referee that the treatment of piezoelectricity is incorrect. The reviewer is right that the piezoelectric tensor for a twisted bilayer may differ significantly than a monolayer, but to the leading order, the piezoelectric effect is the combined response arising from the two individual layer. This is similar to the logic of the response that we provided to reviewer's comment 4 and 8. The so-obtained piezoelectric polarizations by summing up the individual responses have the correct symmetry.

To verify this argument, we have calculated the piezoelectric coefficients for R-type bilayers of different stackings as Fig.R7 shows. The piezoelectric coefficient of the bilayer doubles the value of the monolayer. Since the two layers are aligned, they have the same sign for the piezoelectric coefficient. Therefore, each layer contributes a piezoelectric effect nearly identical to a monolayer.

Fig.R7. The change of areal electric polarization density under a uniaxial strain along x direction for monolayer WSe2, (a), and for various bilayer stackings, (b). A rectangular unit cell geometry is used to calculate the polarizations. In (b), we have divided the total polarization density by the number of layers to compare with the monolayer results in (a).

For the second question, we thank the reviewer for pointing out the difference between the improper and proper definitions of piezoelectric response. The improper piezoelectric coefficient is

$$e_{ijk} = \left(\frac{\partial P_i}{\partial u_{jk}} \right)_{E,T}, \quad (2)$$

which represents the response of the polarization with respect to the strain field. Since the polarization is only defined up to modulo eR/V , the improper definition has a branch dependence. The proper piezoelectric coefficient is defined as [Vanderbilt 1999]

$$\tilde{e}_{ijk} = \left(\frac{\partial J_i}{\partial \dot{u}_{jk}} \right)_{E,T}, \quad (3)$$

which represents the response of the current with respect to the strain flow. The proper definition doesn't depend on the choice of the branch.

According to Ref. [Vanderbilt 1999], the \tilde{e}_{111} component is always the same as the e_{111} component. Under D3h symmetry, the only independent non-zero coefficient is $\tilde{e}_{11} \equiv \tilde{e}_{111}$ and the other coefficients can be obtained as follows:

$$\begin{aligned}\tilde{e}_{122} &= -\tilde{e}_{11}, \\ \tilde{e}_{212} &= \tilde{e}_{221} = -\tilde{e}_{11}.\end{aligned}$$

Therefore, our calculated piezoelectric coefficient follows the proper definition. We have added the relation between the two definitions in the revised Supplemental Materials.

Comment 10: Additionally, even neglecting the piezoelectric polarization, twisted bilayers already have an in-plane polarization arising from the same symmetry breaking which gives rise to the out-of-plane polarization. This is not discussed here, and while I do not think this should affect the Hartree potential either, should be considered alongside the piezoelectric polarization.

Our response: The reviewer probably refers to this paper [Nat. Commun. 14, 1629 (2023)]. The in-plane polarization discussed in the NC paper is a result of stacking difference and is not responsible for the sign flip of Hartree potential drop. To examine its influence on the Hartree potential drop, we conducted calculations for the moiré superlattices without in-plane lattice relaxations but with out-of-plane relaxations. Under this treatment, there are no piezoelectric polarizations, but there are in-plane polarizations discussed in the NC paper and the out-of-plane ferroelectric polarizations. Fig. R8 shows the Hartree potential difference between the two layers at 3.15 degree and 1.70 degree. It can be seen that the position of potential maximum doesn't move with the twist angle. As a result, the in-plane polarizations are not responsible for the change of Chern numbers.

In our paper, the evolution of Hartree potential drop is mainly contributed by piezoelectric potentials which come from the opposite atomic displacements in the two layers. If we take the two layers far away from each other while keep the lattice relaxation unchanged, the polarization discussed in the NC paper will vanish, but the piezoelectric polarization will still exist inside each layer, with their directions opposite.

As the referee suggested, we have cited the NC paper and stated that the polarization discussed in the NC paper does not contribute to the change of band topology,

“Note that additional in-plane polarizations can also arise from the out-of-plane ferroelectricity and local symmetry breaking~\cite{bennett2023polar}. While these effects are present in our DFT calculations, they do not create additional Hartree potential that changes the Chern numbers. ”

Fig.R8. The Hartree potential difference between the two layers for 3.15° and 1.70° tWSe₂ without in-plane relaxations but with out-of-plane relaxations.

Comment 11: The authors develop a MLFF using vasp, then switch to siesta for the DFT calculation. What is the point in training a MLFF in one code, then using it with another code? Why not just train the MLFF using siesta?

Our response: This is mainly due to the balance of computational cost and accuracy. Our large-scale calculations include lattice relaxations and band structure calculations. Compared with VASP, which utilizes a plane-wave basis, SIESTA uses an atomic orbital basis and is therefore less controllable in converging total energy. Hence, for lattice relaxations, we utilize VASP to obtain total energy and the force, which is the derivative of total energy, to train the MLFF.

For band structure calculations, as the interesting frontier bands are mainly composed of low-energy atomic orbitals, plane-wave basis or atomic basis sets should not make much difference. However, it is practically infeasible to calculate moiré bands at small twist angles within VASP. As a result, we resort to using SIESTA for moiré band calculations while using the structures relaxed by MLFF@VASP. To support our argument and validate the accuracy of electronic structures calculated using SIESTA, we conducted comparisons of band structure obtained from VASP and SIESTA in the Supplementary Materials. A good agreement is obtained.

We reproduce the comparison in Fig.R9.

Fig. R9. The comparison between the band structures by using VASP and siesta for 6-degree tWSe₂, (a), and 6-degree tMoTe₂, (b).

Comment 12: I don't understand why the pseudospin skyrmion lattice is mentioned at all here (Fig. 1 and throughout the text). To my knowledge, the pseudospin is a particular interpretation of the Chern number as a winding number (Pontryagin), only for effective Hamiltonians which can be cast in a k .sigma form. Why bother referring to this in a paper about full DFT calculations, which does not contain any effective models? Additionally, the authors use the terms Chern number and skyrmion interchangeably throughout the paper, which would be confusing for a non-specialist.

Our response: The introduction of the skyrmion lattice is intended to demonstrate that for the Chern number to flip sign, the potential extrema must switch between MX and XM. This is also the key idea in our interpretation of the DFT results. Also see our answer to Comment 2. Given the widespread adoption of the continuum model in this field and its likely familiarity to our readers (See Phys. Rev. Lett. 122, 086402 (2019)), we believe that establishing this connection is beneficial. We have improved the description of the Chern number and skyrmion number in the main text to avoid confusion and added a note in the revised manuscript:

“The skyrmion number is defined as

$$N_{sk} \equiv \frac{1}{4\pi} \int_{MUC} dr \frac{\Delta \cdot (\partial_x \Delta \times \partial_y \Delta)}{|\Delta|^3},$$

where the integration is performed within the moiré unit cell. The skyrmion number will flip sign if Δ_z (the inter layer potential drop) flips sign.”

Comment 13: Fig. 2 a is very difficult to read. Consider plotting the displacement and layer separation separately.

Our response: We have modified Fig2 in the revised manuscript as the reviewer suggested.

Comment 14: In Fig. 3 f, the Fourier components of the Hartree potential all seem to become negligible beyond 3. I would have expected them to decay more slowly for decreasing twist angle, when relaxation becomes more significant.

Our response: Although the first Fourier components quickly decay as the twist angles becomes smaller than 3, it should be noted that the relative amplitude among different components changes significantly with decreasing twist angle. The 2nd and 3rd component becomes larger in Fig. 3f, giving similar total potential amplitude at large and small twist angles as shown in Fig. 3e. We reproduce them in Fig.R10.

Fig.R10. The Fourier components of interlayer Hartree potential drop for different twist angles. g_1 is the length of the moiré reciprocal lattice vector.

Comment 15: In Fig. 4 c, lower panel, it looks like C3 symmetry is broken, but this is not the case in any of the others. What is going on here?

Our response: At 1.47 degree, at the gamma point, the C3 symmetry is broken due to the nearly arbitrary mixing of the second and the third band (as these two bands are nearly degenerate). To avoid confusion, we have calculated the bands, Hartree potential, and wave functions at 1.54 degree as shown in Figs.R11-R13, and replaced the 1.47-degree plot with these new plot.

Fig.R11. The band structure of tWSe₂ at 1.54 degree.

Fig.R12. The wave function distribution of the first band at the gamma point for 1.54 degree tWSe₂.

Fig.R13. The DFT Hartree potential drop between the top-layer surface and the bottom-layer surface in 1.54 degree tWSe₂.

Report of Reviewer #2

Comment 1: The manuscript presents multi-scale calculations to accurately investigate the moiré systems from large to small twist angles. Large-scale density-functional-theory (DFT) calculations with machine learning force fields are used for the structure relaxation and moiré band topology. Such systems are attracting significant attention. The authors found that the Chern numbers of the moiré frontier bands change sign as a function of twist angle due to the competition between the in-plane piezoelectricity and the out-of-plane ferroelectricity. This work not only provided a reliable explanation of recent experimental observations in twisted MoTe2 and WSe2 homobilayers. Since the tunable property of a moiré system is sensitive to structural details, an accurate and reliable analysis here is essential. The analysis method could also be extended to other twisted systems. The results of the manuscript are devoted to a hot research topic and are of interest in the field of condensed matter physics. The manuscript can be therefore considered for publication in Nature Communications.

Our Response: We thank the reviewer for praising the importance of our work and recommendation for publication.

Comment 2: As the authors emphasized, the pattern of surface potential changes when the angle is smaller than 1.7° in tWSe2 systems, where the in-plane piezoelectric is relatively enhanced. I am curious on the local structure where the in-plane piezoelectric is large. Whether the local structure is still R-type or H-type stacking? It is recommended that the amplitude and unit of the arrow, namely in-plane displacement field, in Fig. 2(a), and the local structures with maximal displacement should be given.

Our Response: Following the referee's suggestion, we have replotted Fig.2(a) in the revised manuscript and reproduce it in Fig.R14. After checking our relaxed atomic structure, we confirm that the local structure is still R-type as can be seen in Fig.R15. We plot the atomic structures around MM, where maximal displacement occurs in Fig.R15. The approximate position that maximal displacement occurs is labeled. The top view of atomic structures around MM is included in the Supplementary Materials.

Fig.R14. (a), the in-plane displacement field of the top-layer W atoms at 3.15 degree and 1.25 degree for tWSe2. The color and arrow denote the amplitude and direction of in-plane displacement fields, respectively. (b), the inter-layer distance (ILD) distribution at 3.15 degree and 1.25 degree for tWSe2.

Fig.R15. The local atomic structure around the MM site for 3.15-degree, (a), and 1.25-degree tWSe2 relaxed by using neural network potentials. The red dashed circles denote the regions which have the maximal displacement.

Comment 3: The authors generated the training data sets in R-type 6° twisted systems and tested them in 5° systems. Compared to DFT results, the authors claimed the validity of the machine learning model. However, I am concerned about this validity in systems with even smaller twisted angles. If the local stacking is H-type, the systematical comparison would be better via training the model in H-type 6° (namely R-type 54°) twisted data sets.

Our Response: We thank the reviewer for raising this insightful question. In the Supplemental Materials, we have tested the neural network (NN) potential at 6 degree and 5 degree. We further test the NN potential at 3.89 degree for tMoTe2. We summarize the errors of the NN potential in Table R1 for various twist angles. As the twist angle decreases, both the energy RMSE per atom and force RMSE increase, but only slightly and remain negligible compared with convergence threshold. By extrapolating to ~ 1 degree, we estimate the energy RMSE per atom to be less than 5 meV and the force RMSE to be less than 0.07 eV/Å. These errors are still acceptable for the lattice relaxations of moiré superlattice. Therefore, we believe that the NN potential approach remains valid at small twist angles.

We thank the reviewer for the suggestion of training the model. The local stacking is R-type in all our calculations (H type involves flipping of domains which would break bonds). R-type stacking includes MM, MX, and XM high symmetry stacking regions that are represented in both training and testing datasets.

	Energy RMSE per atom	Force RMSE
6°	7.2×10^{-5} eV	0.025 eV/Å
5°	8.3×10^{-4} eV	0.034 eV/Å
3.89°	1.1×10^{-3} eV	0.041 eV/Å

Table R1. The testing energy RMSE per atom and Force RMSE of the NN potential for different twist angles of tMoTe2. The training is done at 6° .

Comment 4: The method details on the surface moiré potential should be given. The amplitude of surface moiré potential presented in this work is somehow smaller than that reported by Jiang Zeng et al. in reference PRB 107, L081402 (2023). In this reference, the authors also reported that the out-plane polarization might not dominate, which coincides with the result of the present work.

Our Response: We thank the reviewer for bringing this paper to our attention. The difference between the surface moiré potential is most likely due to the difference in materials. In addition, we applied coarse-graining techniques to smooth out the atomic-scale oscillations, which reduces the amplitude of the surface electrostatic potential (see our response to comment 3 of reviewer #1). We have added a reference to the above paper in our revised manuscript.

Typo: “angels” in the abstract should be corrected to “angles”

Thanks for catching the typo! Corrected.

Report of Reviewer #3

Comment 1: The authors report on large-scale DFT calculations for twisted bilayers of WSe₂ and MoTe₂ at small twist angles, considering lattice relaxation using molecular dynamics with potentials trained using machine learning plus DFT.

The central result of this paper is as follows: a careful treatment of lattice relaxation provides a more accurate moiré potential, which allowed the authors to reproduce fine experimental results, particularly the Chern numbers of the topologically trivial or non-trivial top valence moiré minibands, and their dependence on twist angle. The authors provide a theoretical explanation to recent experimental results that have established different Chern numbers for the top valence minibands of twisted WSe₂ and MoTe₂; a fact that was not properly explained by continuous models thus far. The authors argue that the different topological invariants for the two materials originate from an inversion of the interlayer electrostatic potential drop at XM and MX domains in the reconstructed moiré superlattice for twisted WSe₂, for twist angles around 1.4°, and which is absent for MoTe₂, at least down to 1.25° according to the authors' calculations.

In my opinion, this paper is both sound and timely, in addition to having direct experimental impact. I therefore recommend its publication in Nat. Commun., once the questions and comments below are properly addressed by the authors:

Our Response: We thank the reviewer for praising the importance of our work and recommendation for publication.

Comment 2: As I mentioned above, the central result of this paper is that lattice reconstruction must be carefully considered in order to obtain an accurate moiré potential. Here, the authors achieve this by relaxing the structures using machine-learning-trained molecular dynamics potentials plus DFT calculations. However, there is a significant body of work by the Manchester theory group, including Refs. [36,37,40,53] of the present manuscript where lattice relaxation is estimated using a different approach (DFT + elasticity theory). It is not clear how and if the authors' approach differs or improves upon the results obtained using the latter mentioned techniques. Can the authors comment on that? In particular, Ref. [53] deals with bilayer WSe₂, which should provide with a suitable point of comparison.

Our Response: We thank the reviewer for the question. We are aware of the vast literature on moiré reconstruction obtained using the elasticity theory or Vincent-Crispi parametrized interatomic potential. In Fig.R16, we compare the interlayer distance of 3.15-degree tWSe₂ from our neural network inferences and the results obtained using the elasticity theory in Ref.[53]. These two different approaches yield qualitatively similar result. However, Ref.[53] didn't show the in-plane relaxations. Comparisons in this aspect cannot be conducted. Such in-plane relaxations are the key to our work.

We believe that the neural network potential holds advantages over the elasticity theory. Firstly, the elasticity theory depends on the choice of model and can be complicated due to parameterization, whereas the neural network potential does not depend on a particular model. The vast size of the deep neural network allows for much higher flexibility to use large amount of training data to improve accuracy and transferability. Secondly, the elasticity theory does not fully account for changes of atomic structures within each unit cell, and is therefore not fully compatible with self-consistent calculations for electronic structures. After all, the NN potential demonstrates accuracy close to full DFT relaxations, but we are not aware of any published work that establish a similar accuracy for the continuum model (please also see our response to comment 3 of reviewer #2). We plan to publish a more systematic comparisons about elasticity theory and Vincent-Crispi parametrized interatomic potential versus the neural network potential in a future work.

Fig. R16. The interlayer distance distribution of 3-degree tWSe2 obtained using the neural network, (a), and the elasticity theory, (b).

Comment 3: I believe the work by Pan et al. on the band topology of twisted WSe2 bilayers from 2020 merits citation: [Phys. Rev. Research 2, 033087 (2020)]. This should also nicely complement the comparisons with Ref. [20], which studied the case of MoTe2. Both references use continuous models limited to the first star of moiré Bragg vectors (neglect reconstruction).

Our Response: We have cited this paper in the revised manuscript.

Comment 4: In line 127 of the manuscript, the authors mention a computational-cost-saving measure which I, as a non DFT specialist, did not understand even after reading Supplementary Sec. IV. The authors mention using both SIESTA and VASP, which sounds like double the work. It seems to me that one package deals with SOI while the other doesn't, but I would prefer it if the authors themselves could clarify this point, especially in the main text.

Our response: Please see our response to comment 11 of reviewer #1. We have added more details in the Supplementary Materials and modified the sentences in line 132-139 in the main text as follows,

“To reduce the computational cost, we adopt the `\textsc{siesta}` package~\cite{soler2002siesta} for band structure calculations. We first benchmark the accuracy of this local basis approach with the plane-wave basis approach by comparing the band structures at 6° obtained from `\textsc{siesta}` and `\textsc{vasp}`, and reach a qualitative agreement between the two (see details in Ref.\cite{supp}). Then we perform small twist-angle band calculations by using the `\textsc{siesta}` package.”

Comment 5: Likewise, the comment at line 140 about introducing an artificial Zeeman splitting between the two valleys is not entirely clear to me. Was the purpose of this to minimize intervalley mixing by the moiré potential? If so, how large was the Zeeman splitting introduced, and how does it compare with intervalley mixing terms? I could not find this discussion in the Supplementary Materials.

Our response: This is a specific technique in DFT. In order to calculate the valley Chern number, we need to separate bands from the two valleys. Hence, a small Zeeman field (0.1 meV) is added.

Comment 6: Finally, in line 212 the authors state that "The intricate behavior of the surface potential goes beyond the continuum approximation of moiré potential based on the first-star expansion of the reciprocal lattice vectors alone (...)", with which I agree. Although Ref. [20] and my strongly suggested reference [Phys. Rev. Research 2, 033087 (2020)] do stop at first stars of moiré vectors, Ref. [53] of the present manuscript does formulate multiple continuous models (for R and H cases and for different points in the Brillouin zone) that incorporate lattice reconstruction, and which include multiple stars of Bragg vectors. Indeed, Fig. 16b of that reference matches the spatial distribution of holes reported by the authors in Fig. 4c for about the same angle. Can the authors comment on how those results compare with theirs, and on

whether they believe that, once the reconstruction has been properly calculated, the results of the present manuscript could be reproduced by a continuous model that incorporates that reconstruction? I believe the authors' input on this matter will be valuable to readers.

Our response: The band structures and band edge distribution presented in Ref. [53] do not fully align with our results. Firstly, in Fig.12(a) of Ref.[53], at 3.0 degree, the valence band edge is at MM, whereas our calculations indicate that the valence band edge is at MX/XM (see Fig.4). Secondly, in Fig.16(b) of Ref.[53], at 1.4 degree, the separation between the first band and the second band is around 13 meV, whereas our calculations show that at 1.25-1.54 degree, the separation between the first band and the second band is less than 5 meV. All these features are important starting point to accurately model correlated topology phases of moiré bilayers.

These discrepancies between our DFT band results and continuum model calculations of Ref.[53] are likely caused by the structure details and the incomplete treatment of lattice relaxation effects in continuum models. For example, there could be differences for the in-plane relaxations between the elasticity theory and the neural network inferences. On the other hand, in Ref.[53], the in-plane lattice relaxations effects are not fully included for the K-valley moiré bands. Such in-plane lattice relaxations are important, considering the formation of domain walls at small twist angles, which can significantly reshape the moiré potential and interlayer tunneling. In addition, in Ref.[53], the strain-induced scalar and vector potentials are not included. To address these issues, we are currently developing continuum models using the moiré superlattice obtained from the neural network. While the method largely follows the same spirit as Ref. [53], at this stage, we feel there are too many subtleties to us to comment.

Reviewers' Comments:

Reviewer #1:

Remarks to the Author:

Thanks to the authors for addressing my (many) comments well. I think some of the key aspects of this work weren't explained well enough in the original manuscript (like the piezoelectric polarization, and why changes in the polarization would lead to changes in Chern numbers), but they have been cleared up now. I think this work can be published now.

I have a few more suggestions which may optionally be addressed:

- Regarding the MLFF which has been trained with VASP and used with SIESTA: I'm curious to know if the authors used a good basis set in SIESTA which is converged with respect to plane wave calculations. It is well-known that the default basis sets do not always work well, especially for 2D materials. Lattice constants can be wrong by a few %. I would be interested to see what the difference in lattice constant is for a single layer of MoTe₂, using VASP with the parameters used to train the MLFF, and using siesta, with the basis set used in the calculations. If the difference is comparable to the strain which results in piezoelectricity, then that would call into question whether the effects observed in this work are realistic or an artefact of the calculations.

- Another (more expensive) way to check how robust the results are would be to perform single point calculations with VASP and calculate the Hartree potential.

- I have to disagree with the response to my comment 7. Just because a certain terminology has been used in other papers, potentially in high impact journals, that does not mean that they are correct. Using a particular nomenclature because other people have used it in previous works, or for convenience, is not a good excuse if it is not a correct description of the physics. The authors did agree that ferroelectricity is not an appropriate term to use here, and I would recommend not using the term so loosely in this work. They want to make a distinction between two separate mechanisms for polarization, fine. But they shouldn't call one of them something they know is not correct because it is easier. One is coming from an interlayer electronic charge transfer due to the symmetry breaking of different stackings (which results in ferroelectricity in commensurate layers, but not here), and one is coming from an equal and opposite strain on the different layers, resulting in an additional interlayer dipole. I think it is possible to distinguish between the two effects without being misleading.

Reviewer #2:

Remarks to the Author:

The authors modified and enhanced the manuscript after addressing questions from the referees. I would recommend its publication in Nature Communications.

Reviewer #3:

Remarks to the Author:

I am satisfied with the authors' responses to my queries, and those of the other referees. I reiterate my recommendation to publish.

Response to Reviewers' comments

We greatly appreciate the reviewers' feedback. Below, we respond to their comments in a point-by-point format.

Report of Review #1

Comment 1: Thanks to the authors for addressing my (many) comments well. I think some of the key aspects of this work weren't explained well enough in the original manuscript (like the piezoelectric polarization, and why changes in the polarization would lead to changes in Chern numbers), but they have been cleared up now. I think this work can be published now.

Our Response: We thank the reviewer's recommendation for publication.

Comment 2: I have a few more suggestions which may optionally be addressed:

Regarding the MLFF which has been trained with VASP and used with SIESTA: I'm curious to know if the authors used a good basis set in SIESTA which is converged with respect to plane wave calculations. It is well-known that the default basis sets do not always work well, especially for 2D materials. Lattice constants can be wrong by a few %. I would be interested to see what the difference in lattice constant is for a single layer of MoTe₂, using VASP with the parameters used to train the MLFF, and using siesta, with the basis set used in the calculations. If the difference is comparable to the strain which results in piezoelectricity, then that would call into question whether the effects observed in this work are realistic or an artefact of the calculations.

Another (more expensive) way to check how robust the results are would be to perform single point calculations with VASP and calculate the Hartree potential.

Our Response: In our study, we used the double-zeta plus polarization basis within SIESTA. The relaxed lattice constant of monolayer MoTe₂ is 3.527 angstroms using VASP and 3.520 angstroms using SIESTA. The difference between the two is ~0.2%. Given the well-known inaccuracies associated with DFT calculations of lattice constants, we used experimental lattice constants, i.e. 3.495 angstrom for MoTe₂ and 3.28 angstrom for WSe₂. It's important to note that variations in lattice constants will not impact the piezoelectricity results qualitatively. This is because the piezoelectric charge density is directly proportional to the strain gradient.

Comment 3: I have to disagree with the response to my comment 7. Just because a certain terminology has been used in other papers, potentially in high impact journals, that does not mean that they are correct. Using a particular nomenclature because other people have used it in previous works, or for convenience, is not a good excuse if it is not a correct description of the physics. The authors did agree that ferroelectricity is not an appropriate term to use here, and I would recommend not using the term so loosely in this work. They want to make a distinction between two separate mechanisms for polarization, fine. But they shouldn't call one of them something they know is not correct because it is easier. One is coming from an interlayer electronic charge transfer due to the symmetry breaking of different stackings (which results in ferroelectricity in commensurate layers, but not here), and one is coming from an equal and opposite strain on the different layers, resulting in an additional interlayer dipole. I think it is possible to distinguish between the two effects without being misleading.

Our Response: We agree with the reviewer that there is ambiguity in the particular nomenclature that describes the effect coming from asymmetric interlayer coupling and the polarization pattern in the moiré system. However, we still believe that “moiré ferroelectricity” is among the appropriate names that describe this particular effect, due to the fact that this nomenclature has been used widely in the moiré community.

To reflect this discussion, we have modified our manuscript. The relevant sentence now reads “The ferroelectric effects arise from the inversion symmetry breaking in *R*-type TMD bilayers, and have been termed ‘moiré ferroelectricity’.”